# PLP-RC: Point–Line–Plane Fusion for Discriminative Relation Classification with LLMs

## Abstract

Relation classification is a fundamental NLP task that involves identifying the semantic relations between entity pairs in a given text. While pre-trained language models have advanced this area, effectively integrating local entity information with global context remains a key challenge. Large Language Models offer rich world knowledge, but their generative use often suffers from hallucinations, limiting reliability. To address these issues, we propose a Point–Line–Plane fusion framework for discriminative relation classification with LLM embeddings. Entity spans are modeled as local point representations, the end of sequence token provides a global plane representation, and an attention-based line representation aligns the two. This discriminative paradigm avoids hallucinations while fully exploiting LLM representations. Our method achieves new SOTA performance on TACRED, TACREV, and RE-TACRED benchmarks, outperforming both discriminative and generative baselines. Ablation studies provide further evidence for the effectiveness of our design in achieving context-aware relation classification.

## 1 Introduction

Entity Relation Extraction (RE) is one of the fundamental tasks in natural language processing (NLP), which aims to identify entities from unstructured text and determine the semantic relations among them, ultimately producing structured triples in the form of (subject, relation, object) (Zhao et al. (2024)). RE is essential for numerous downstream applications, including knowledge graph construction, question answering, information retrieval, and graph-augmented generation. Within this context, relation classification plays a central role, as it determines the semantic relation between entity pairs and directly impacts the accuracy and completeness of the extracted triples. Hence, its effectiveness is critical for the reliability of downstream applications and the overall utility of RE (Han et al. (2025)).

The evolution of relation classification reflects a systematic shift in modeling entities and their contexts. Early approaches relied on rule-based (Kambhatla (2004)) and feature-engineering methods , leveraging syntactic templates and statistical co-occurrence patterns. These methods, however, depended heavily on expert knowledge and domain-specific linguistic resources, limiting their generalization and portability. The advent of representation learning marked a paradigm shift. Neural networks enabled more effective modeling of semantic interactions between entities and context through distributed embeddings. This approach was further advanced by pre-trained language models (PLMs) based on the Transformer architecture, such as BERT (Devlin et al. (2019)) and its variants, which produce rich contextualized representations and achieve strong performance. Despite their success, even models like SpanBert (Joshi et al. (2020))struggle with long-range dependencies and discourse-level reasoning required for cross-sentence relation extraction.

In recent years, large language models (LLMs) (Wang et al. (2022)) and frameworks such as retrieval-augmented generation (RAG) (Efeoglu & Paschke (2024)) have been introduced to handle complex relational structures and challenges like long-tail relations, semantic ambiguity, and relation overlap. Despite their notable progress, these approaches also introduce new issues, including high computational costs and a strong susceptibility to hallucination (Huang et al. (2025)). Due to the inherent limitations of generative paradigms, existing generation-based methods struggle to effectively leverage LLM knowledge for relation classification without triggering hallucination.

Meanwhile, although BERT and LLMs still exhibit fundamental limitations in capturing global contextual semantics. These limitations arise from their distinct pre-training objectives—Masked Language Modeling for BERT and next-token prediction for LLMs—and are further exacerbated in LLMs by causal attention, which restricts access to full bidirectional context (Yin et al. (2024)). Consequently, the token representations often lack the holistic semantic structure required for relation classification and other complex semantic understanding tasks.

These limitations undermine the robustness and generalizability of relation classification systems. To address these challenges, we propose a unified framework that integrates discriminative modeling, a geometry-inspired contextual fusion mechanism, and prompt-based instruction enhancement:

- **Discriminative Model Based on LLMs**: We leverage the comprehensive and domain-specific knowledge of LLMs to overcome the limitations of traditional models. To prevent hallucination, we utilize LLMs as powerful feature encoders within a discriminative framework, ensuring both broad knowledge coverage and reliable output.
- **'Point–Line–Plane' Contextual Fusion Mechanism**: We conceptualize the model in terms of a geometric analogue: tokens are treated as points, their relationships as lines (as captured by attention weights), and the entire context as a plane embodied by the [EOS] token.
- **Instruction-Enhanced Method Based on Prompt Learning**: We augment the model's ability to capture task-aware semantics by appending carefully designed natural language instructions to the input sequence. These instructions explicitly guide the model toward relation classification, strengthening the representation of the [EOS] token as a global contextual anchor.

Taken together, our framework provides a principled and robust solution to relation classification, offering improved interpretability and generalization, and achieves new state-of-the-art performance among discriminative approaches on TACRED, TACREV, and RE-TACRED.

## 2 RELATED WORK

### 2.1 RELATION CLASSIFICATION

Relation extraction (RE) is a core task in information extraction, aiming to identify entities (e.g., persons, locations, organizations) and the semantic relations between them. Early approaches such as SpanBert (Joshi et al. (2020)) leverage PLMs to encode spans and adopt discriminative methods for relation classification, achieving strong performance. Subsequent methods (Lyu & Chen (2021)) further incorporate entity type information to constrain candidate relations, improving classification accuracy. However, these approaches primarily rely on point or span-based representations, or additional constraints, without fully exploiting contextual cues and the knowledge encoded in pretrained models. More recently, LLM-based generative methods, such as DeepStruct (Wang et al. (2022)), pretrain on large-scale task-agnostic corpora and perform zero-shot inference, achieving state-of-the-art results but at the cost of massive training resources, high inference overhead, and increased hallucination risk. Retrieval-augmented approaches such as RAGRE (Efeoglu & Paschke (2024)) and RAGRE+Finetuned (Efeoglu & Paschke (2025)) improve reliability by injecting external knowledge, yet suffer from dependency on retrieval modules and limited generalization. These limitations motivate our work, where we propose a Point–Line–Plane fusion framework to better integrate entity and contextual information for relation extraction.

### 2.2 LLM EMBEDDING

Large language models (LLMs), pretrained on massive text corpora, have become fundamental for a wide range of NLP tasks. Recent work such as GTE (Li et al. (2023)), GME (Zhang et al. (2024)), and E5 (Wang et al. (2024)) demonstrates that compact embeddings derived from LLMs can effectively support applications like retrieval and classification. These approaches typically rely on the final hidden state of Transformers, with the last token serving as the sequence representation.

Unlike generative paradigms that model the full data distribution, embedding-based methods follow a discriminative paradigm. By focusing explicitly on the decision boundary rather than text generation, discriminative methods are computationally more efficient and substantially less prone to hallucinations. Prompt-based methods such as BGE emb (Xiao et al. (2023)) further enrich embeddings

with task-specific natural language instructions, showing that instruction-driven representations can significantly enhance downstream performance. These observations motivate our work to adopt a discriminative, embedding-based approach for relation classification with LLMs.

## 3 PRELIMINARY

### 3.1 HALLUCINATION ANALYSIS OF GENERATIVE METHODS

LLMs often generate responses that deviate from user input or training data, a phenomenon known as "hallucination" (Bang et al. (2025)). The hallucination issues is mainly attributed to two factors: the next-token prediction objective used during pretraining, and the low quality of the pretraining corpus. Existing efforts to mitigate hallucination focus on improving data quality and training objectives, or incorporating external knowledge at inference time through methods such as RAG and constrained decoding. In contrast, discriminative paradigms inherently avoid hallucination by operating over predefined candidate spaces, offering a more stable alternative to generative approaches.

### 3.2 TASK DEFINATION AND MOTIVATION

Given an input sequence $X = (x_1, x_2, ..., x_n)$, the goal of the relation classification task is to predict the semantic relation between a subject entity and an object entity within the input sequence $X$, which can be formalized as Equation 1.

$$\hat{r} = \arg \max_{r \in \mathcal{R}} P(r \mid X) \tag{1}$$

We define $\mathcal{R}$ as the set of candidate relation types, with $r \in R$ denoting a specific type. Following a span-based discriminative paradigm, the training objective can be formulated as maximizing the Score of the correct relation $r^*$ as Equation 2:

$$\hat{\theta}, \hat{\phi}, \hat{\gamma} = \arg \max_{\theta, \phi, \gamma} S_\gamma(r^*, f_\phi(H_{\text{sub}} \mid X, \theta), f_\phi(H_{\text{obj}} \mid X, \theta)) \tag{2}$$

where $H = \text{PLM}(X; \theta)$ denotes the full hidden states of the input sequence $X$ produced by the PLM with parameters $\theta$, and $H_{\text{subject}} = (h_{s_1}, h_{s_2}, \ldots, h_{s_m}), H_{\text{object}} = (h_{o_1}, h_{o_2}, \ldots, h_{o_n}) \subseteq H$ are the subsequences corresponding to the subject and object entity spans. $f_\phi$ is a span-level fusion network that maps an entity span $H_{\text{entity}}$ to a span-level feature vector, with $\phi$ denoting its parameters. $S$ is a scoring function with learnable parameters $\gamma$ that measures how well the predicted relation aligns with the ground-truth label $r^*$. In short, this objective maximizes the score of the correct relation with respect to the PLM parameters $\theta$, the fusion network parameters $\phi$ and the scoring function module parameters $\gamma$.

While encoder-only PLMs such as BERT benefit from bidirectional attention and provide context-sensitive embeddings, their token representation is not explicitly optimized for relation classification, as the masked language modeling pretraining objective does not enforce span-level semantic coherence. In contrast, decoder-only LLMs are pretrained with the objective of next-token prediction in Equation 3,

$$\mathcal{L}(\theta) = -\sum_{t=1}^{T} \log P(x_t \mid x_{<t}; \theta) \tag{3}$$

where $T$ denotes the length of the training sequence, t indexes each position in the sequence, and the model predicts each token given only the preceding context $x_{<t}$. This objective favors autoregressive fluency but lacks explicit mechanisms for learning coherent span-level semantics. Consequently, both paradigms have inherent limitations when directly adapted for relation classification, even with task-specific fine-tuning. To address this limitation, we integrate the global context representation with span embeddings, and define the prediction objective as Equation 4.

$$\hat{\theta}, \hat{\phi}, \hat{\psi}, \hat{\gamma} = \arg \max_{\theta, \phi, \psi, \gamma} S_\gamma(r^*, f_\phi(H_{\text{sub}}|X, \theta), f_\phi(H_{\text{obj}}|X, \theta), g_\psi(H|X, \theta)) \tag{4}$$

where $g_\psi$ denotes the global context that produces a representation of the full hidden states $H$, with $\psi$ donates the global context fusion network's parameters, capturing long-range dependencies and context information beyond individual entity spans. By integrating context-aware features with span features, the model captures richer semantic signals and overcomes the limitations of existing PLMs.

## 4 METHODOLOGY

Effective relation classification requires modeling global context beyond entity mentions. In decoder-only LLMs, the [EOS] token is commonly used as a compressed representation of the entire input (Springer et al. (2024)), and it is derived through causal attention, which compresses preceding tokens in a strictly autoregressive manner. This aggregation produces a compact global signal but inevitably flattens the underlying sequence topology and discards fine-grained token interactions.

Building on this motivation, we propose the Point-Line-Plane framework for relation classification (PLP-RC). Unlike generative paradigms, our discriminative formulation leverages the rich knowledge embedded in pretrained LLMs while avoiding hallucinations. Rather than relying on a single global summary, PLP-RC aggregates cues from multiple levels of granularity, including information from entities, the surrounding context, and direct interaction patterns between them. The goal is to construct a holistic representation that captures not only the entities and their context but also the explicit relationship between them.

To better capture relational cues, we decompose the global semantic information into three complementary components: (1). The Point representation serves as a localized coordinate, embedding the fine-grained semantic content of an entity span. (2). The Plane representation, derived from the [EOS] token's final hidden state, acts as a low-dimensional projection of the global contextual manifold. (3). Crucially, the Line representation bridges these two levels of granularity. Figure 1 illustrates the overall architecture.

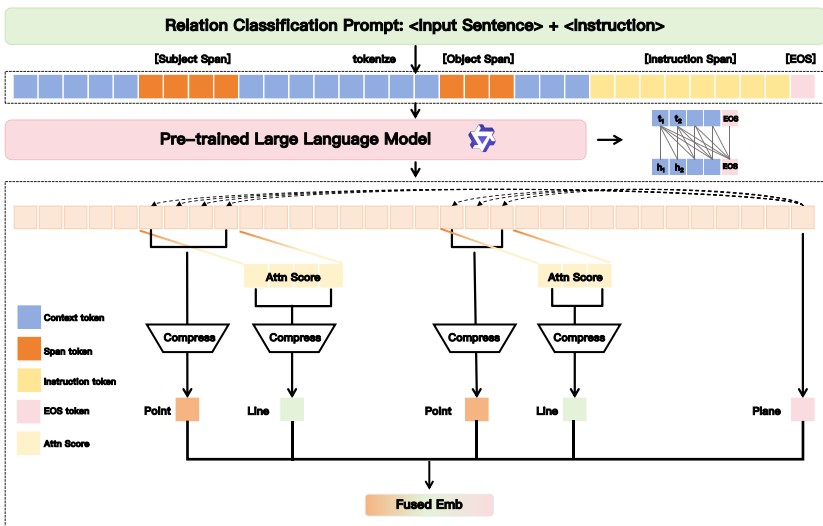

Figure 1: Overall architecture of the proposed framework. The Point–Line–Plane fusion module constructs point embeddings from entity spans, line embeddings from attention-based correlations between entities and the [EOS] token, and plane embeddings from the global contextual representation. These features are integrated into fused entity representations for relation prediction. The prompt-based instruction module further guides the decoder-only LLM with task-specific prompts to enrich contextual semantics.

### 4.1 POINT-LINE-PLANE FUSION METHOD

We employ the decoder-only LLMs as our text encoder to obtain contextualized token-level representations of the input sequence. Compared with encoder-only models such as BERT, decoder-only LLMs pretrained on large-scale corpora demonstrate superior scalability and stronger contextual modeling capacity, making them better suited for constructing span-level and context-aware features in our fusion framework. Building on these representations, we construct point, line, and plane features to jointly capture local entity information, relational interactions, and global contextual semantics, respectively.

### 4.1.1 POINT: DIRECT SPAN-LEVEL REPRESENTATION

In the task of relation classification, the most fundamental signal arises from the entity mentions themselves. Given an encoded sequence $H$, each entity is represented either by a single token embedding or by a contiguous span of embeddings $H_{\text{entity}} \subseteq H$. Such spans provide the most direct semantic evidence for relation prediction, since they capture the lexical and local contextual information tied to the entity surface form. To transform the span into representations, several strategies have been explored in prior work, such as mean pooling, max pooling, or concatenating the boundary token embeddings (Fu et al. (2021)). We refer to these span-level entity embeddings as point features. Specifically, we construct Point representations by combining boundary embeddings with entity type information. Leveraging the contextualized token embeddings from decoder-only LLMs, we construct span representations by integrating boundary embeddings through dedicated MLP layers. To enrich entity semantics, we further embed discrete entity types, mapping one-hot type encodings into continuous vectors $e_{\text{type}}$ and fused with span embeddings in Equation 5.

$$e_{\text{Point}} = \text{MLP}_{\text{start}}(h_{\text{start}}) + \text{MLP}_{\text{end}}(h_{\text{end}}) + e_{\text{type}} \in \mathbb{R}^d, \tag{5}$$

where boundary start and end tokens are projected independently via dedicated MLP layers, and combined with type embeddings to form the final span representation. Although token sequences and point embeddings encode partial contextual cues, they remain insufficient for capturing relational semantics, partly due to the limitations of the pretraining objectives in Equation 3, which do not explicitly optimize for entity interactions. This motivates the incorporation of higher-level features to better capture relational information.

### 4.1.2 PLANE: THE COMPRESSED REPRESENTATION OF THE CONTEXT

Context is critical for relation classification, as the same entity pair may imply different relations under varying contexts. Relying solely on entity spans limits generalization. Intuitively, the global compressed representation should integrate point-level embeddings together with the structural and topological relations between them, providing the semantic representation. However, performing additional weighted summations would lead to computational efficiency issues. Instead, by leveraging the characteristics of PLMs, we can obtain compressed contextual representations more efficiently.

Due to the causal attention mechanism, information is aggregated and propagated forward, and the [EOS] token is commonly used as the compressed representation of the whole input. In line with prior work, we take the [EOS] as the global representation of the entire sequence, denoted as $e_{\text{Plane}} = h_{EOS} \in \mathbb{R}^d$. While the [EOS] provides a compact summary of the entire sequence, its computation via attention-weighted pooling limits its ability to model fine-grained interactions between entities and context. To bridge this gap, we further exploit attention scores to quantify the correlation between [EOS] and entity boundaries, yielding the line representation that bridges local entity information and the global context.

### 4.1.3 LINE: THE ASSOCIATION BETWEEN TOKENS

In causal attention mechanism, the attention score still reflects the degree of association between tokens, but the computation is constrained by the autoregressive mask that prevents each position from attending to future tokens. The computation of attention score is computed as Equation 6

$$A = \text{Softmax}\left(\frac{QK^\top}{\sqrt{d_k}} + M\right) \tag{6}$$

where $Q \in \mathbb{R}^{n \times d_k}$ denotes the query, $K \in \mathbb{R}^{n \times d_k}$ denotes the key, and $d_k$ is the dimension of the keys. The matrix $M \in \mathbb{R}^{n \times n}$ is a causal mask that assigns $-\infty$ to positions corresponding to future tokens, thereby enforcing that each token can only attend to itself and its preceding context. To enable better interaction between the local span representation and the [EOS] token that encodes global contextual information, we adopt the attention score to extract their correlations as the line information. Line information serves as an implicit edge representation, analogous to edge weights in GNNs (Zhou et al. (2020)), enabling structured message passing between local entity spans and the global [EOS] context. Specifically, we compute the attention scores between the [EOS] token and the entity boundaries (i.e., entity start and entity end), as well as the average attention score over

all tokens within the entity span to capture aggregated interactions. These three components are then concatenated to form the line representation $e_{\text{Line}}$ in Equation 7:

$$e_{\text{Line}} = \text{Concat}(A_{\text{start}}, A_{\text{end}}, \frac{1}{N_t} \sum_{i=\text{head}}^{\text{end}} A_i) \tag{7}$$

where $A_i$ denotes the attention score between token $t_i$ and [EOS], and $N_t$ is the number of tokens in the entity span. In this way, we obtain the point information $e_{\text{Point}}$, the line information $e_{\text{Line}}$, and the plane information $e_{\text{Plane}}$. The line embedding $e_{\text{Line}}$, derived from attention scores, has a relatively small dimensionality (equal to 3 times the number of attention heads). To amplify its contribution, we project it into the same latent space through an $\text{MLP}_A$. At the entity level, we enrich point representations by element-wise addition with the transformed line embedding. Following the intuition of positional encoding, we incorporate the transformed line embedding into the entity representation by element-wise addition, as formalized in Equation 8.

$$e_{\text{Entity}} = e_{\text{Point}} + \text{MLP}_A(e_{\text{Line}}) \tag{8}$$

which ensures that line information acts as a bias term enriching the entity representation without overriding the semantics of the point embedding. Besides, the plane information Plane encodes complementary high-level structural semantics rather than fine-grained entity attributes. To preserve the independent contributions of heterogeneous sources, we adopt concatenation in Equation 9:

$$e_{\text{Fused}} = \text{Concat}(e_{\text{Plane}}, e_{\text{Sub}}, e_{\text{Obj}}) \in \mathbb{R}^{3 \times d} \tag{9}$$

where $e_{\text{Subj}}$ and $e_{\text{Obj}}$ are the entity embeddings obtained from the above fusion process. A linear classifier followed by a softmax layer produces the predictive distribution over the relation label set $r$, as defined in Equation 10.

$$\mathcal{L} = -\frac{1}{N} \sum_{i=1}^{N} \log p(r^i \mid e_{\text{fused}}^i), \tag{10}$$

Here, $N$ denotes the total number of examples. This design maintains the full expressive power of each component while allowing the classifier to learn their relative importance during training.

### 4.2 INSTRUCTION REFINEMENT BASED ON PROMPT LEARNING

Since the optimization objective of LLMs during pre-training is next-token prediction, whereas in practical applications we use the last hidden state to represent the overall context, there exists a semantic gap between the two. To enable the [EOS] to sufficiently capture contextual information for entity relation classification, we enhance its semantic representation through instruction tuning.

Specifically, we design a relation classification instruction $I$, constructed in natural language using the following template. This instruction is concatenated to the end of the original input context, i.e., $C' = C + I$. In this way, we aim to strengthen the task-aware contextual representation provided by the [EOS] token. The construction of the relation classification instruction is illustrated in 2.

> **Prompt**
>
> **Context**: U.S. District Court Judge Jeffrey White in mid-February issued an injunction against Wikileaks after the Zurich-based Bank Julius Baer accused the site of posting sensitive account information stolen by a disgruntled former employee.
> **Subject:** Julius Baer
> **Object:** Jeffrey White
> **Instruction:** Predict the relation between the entity pair [Subject] and [Object] :
> **Context' = Context + Instruction**

Figure 2: Construction of the relation classification instruction appended to the context.

## 5 EXPERIMENTS

### 5.1 DATASET

We adopt TACRED and two revised versions with corrected labels TACREV and RE-TACRED as the benchmarks in this paper. TACRED is a large-scale, comprehensive, and task-oriented super-

vised relation extraction dataset that significantly outperforms previous datasets, which are either limited in size or heavily affected by noise. These datasets are among the most widely used benchmarks for supervised relation classification, providing a reliable and comprehensive evaluation foundation for this task.

- TACRED (Zhang et al. (2017)) (The TAC Relation Extraction Dataset) is a supervised relation extraction dataset created through crowdsourcing, specifically designed for TAC KBP relations. The relations are not pre-assigned with directions, meaning they can be extracted from sentence tokens.

- TACREV (Alt et al. (2020)) is a refined version of TACRED. This revised dataset addresses the issues of data annotation quality and relation ambiguity, which constitute the primary bottlenecks to model performance on TACRED.

- RE-TACRED (Stoica et al. (2021)) is a re-annotated version of TACRED that enables more reliable evaluation of relation extraction models.

By leveraging these datasets, which have become the widely adopted standard for relation classification research, we ensure that our evaluation is both rigorous and comparable to prior work. More dataset details can be found in the Appendix.

## 5.2 BASELINE

We consider the following baselines for comparison and we divide the baselines into two categories: discriminative Conventional Models methods and generative large language model (LLM) methods. **Conventional Model Methods** : PALSTM (Zhang et al. (2017)), C-GCN (Zhang et al. (2018)), SpanBert (Joshi et al. (2020)), KnowBERT (Peters et al. (2019)),LUKE (Yamada et al. (2020)), Roberta (Zhou & Chen (2022)); **LLM-based methods** : DeepStruct (Wang et al. (2022)) , GAP (Chen et al. (2024)), RAGRE (Efeoglu & Paschke (2024)), RAGRE+Fintuned (Efeoglu & Paschke (2025)) . We demonstrate the effectiveness of our proposed method through extensive experiments comparing it with the aforementioned baseline approaches.

We restrict our comparisons to open-source models, spanning both discriminative and generative paradigms, as closed-source models have accessibility and transparency limitations, ensuring a fair and representative evaluation.

## 5.3 IMPLEMENTATION DETAILS

Our model consists of two components: a LLM encoder and a feature fusion-based relation classifier. We take dense type Qwen3 Series as our pretrained LLM backbone. We take models ranging from 0.6B to 4B parameters for our experiments. In addition, our relation classification module consists of a feature extraction and fusion component that captures the PLP-RC representations of entities, followed by a two-layer MLP with ReLU activation functions for relation classification.

In the training stage, we conduct an end-to-end training with instructions to adapt to the downstream relation classification task with full parameter tuning. All experiments can be conducted on a single NVIDIA H100 GPU 80GB. More hyperparameter settings can be found in the Appendix.

For a fair comparison with previous methods, we also use micro-F1, a commonly adopted metric in multi-class classification tasks especially entity relation classification tasks, as our evaluation metric.

## 5.4 EXPERIMENT RESULT

Table 1 summarizes the overall micro-F1 results across TACRED, TACREV, and Re-TACRED. Our Qwen3-0.6B model, enhanced with the PLP-RC method, achieves a micro-F1 of **88.9** on TACRED and **92.8** on TACREV, outperforming all prior pre-trained and generative large language model baselines by a clear margin. Notably, despite its relatively small size, the 0.6B model even surpasses much larger models such as Flan-T5-XL and LLaMA2-7B. This demonstrates that the discriminative paradigm, by focusing directly on decision boundaries rather than sequence generation, can deliver superior task alignment and sample efficiency.

On Re-TACRED, our method attains **91.1**, which is on par with the best existing approaches (e.g., Roberta+Typed and GAP), showing that the proposed PLP-RC representation does not compromise robustness across benchmark variants. Furthermore, scaling to larger Qwen3 models (1.7B and 4B) yields additional performance gains, establishing new state-of-the-art results among decoder-only generative LLMs on all three benchmarks. Overall, these findings highlight the advantages of reframing LLMs as discriminative classifiers: smaller models achieve superior task alignment compared to larger generative counterparts, while larger models continue to benefit from scaling, delivering both efficiency and competitiveness.

| Method | Model | TACRED | TACREV | RE-TACRED |
|---|---|---|---|---|
| *Sequence-based Model* | | | | |
| PA-LSTM | LSTM | 66.2 | 74.3 | 79.4 |
| C-GCN | GCN | 66.7 | 75.0 | 80.2 |
| *Transformer-based (Large) Language Model* | | | | |
| SpanBert | Bert | 66.3 | 73.4 | 83.2 |
| KnowBert | Bert | 71.5 | 79.3 | - |
| LUKE | Bert | 72.7 | 80.6 | 90.3 |
| Bert + Typed | Bert | 72.9 | 81.3 | 89.7 |
| Roberta + Typed | Roberta | 74.6 | 83.2 | 91.1 |
| GAP | Roberta | 72.7 | 82.7 | 91.4 |
| RAGRE | Flan-T5-XL | 86.6 | 88.3 | 73.3 |
| *Decoder-Only Large Language Model* | | | | |
| DeepStruct | GLM-10B | 76.8 | - | - |
| RAGRE + Fintune | LlaMA2-7B | 84.5 | 90.2 | 75.1 |
| RAGRE + Fintune | Mistral-7B | 84.7 | 87.5 | 88.3 |
| *Ours* | | | | |
| PLP-RC | Qwen3-0.6B | 88.9 | 92.8 | 91.1 |
| PLP-RC | Qwen3-1.7B | 89.4 | 93.4 | 92.4 |
| PLP-RC | Qwen3-4B | **89.9** | **94.0** | **92.9** |

Table 1: Overall Micro-F1 results on TACRED, TACREV, and Re-TACRED, comparing our PLP-RC method with prior pre-trained and decoder-only generative LLM methods.

## 5.5 ABLATION STUDY

We conduct comprehensive ablation experiments to validate the effectiveness of our method. The details of ablation study are as follows.

**Ablation of PLP-RC components**: We perform ablation studies to examine the effect of each component in our method. We sequentially remove point and line information to examine their individual contributions. The results are as shown in Figure 3a.

These results highlight the importance of each component in our framework. The complete PLP-RC framework achieves the best performance across all datasets. While removing Line information alone only leads to a minor drop, removing both Line and Plane substantially degrades performance, suggesting that Plane plays a more critical role and that Line information is more effective in combination with Plane. While Point captures fine-grained entity semantics, Line and Plane serve as essential components for modeling global context. Moreover, removing Instruction results in the largest performance degradation, confirming its strong influence on the overall effectiveness. Overall, the ablation confirms that each component contributes to the effectiveness of PLP-RC. The overall results suggest that PLP-RC forms a solid representation backbone, upon which instruction can further enhance performance.

**Ablation of model parameters scaling law**: We take Qwen series model range from 0.5B to 4B in our model parameters ablation experiments. For 0.5B model, we choose Qwen2.5-0.5B as our LLM backbone. The results are shown in the Figure 3b and Table 1. Scaling up model parameters consistently enhances benchmark performance. We also observe that although increasing the model parameters leads to further improvements in performance, the gains are non-linear with respect to the scale of parameter growth. The performance gain from using larger models is limited under

the discriminative setting, and does not align with trends observed in Qwen3 Technical Report. This suggests that pure generative capability does not fully reflect effectiveness on non-generative downstream tasks such as relation classification.

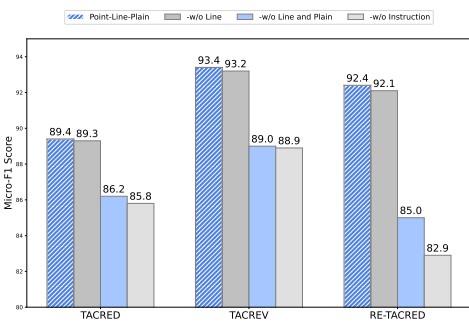

(a) Ablation Results of Components

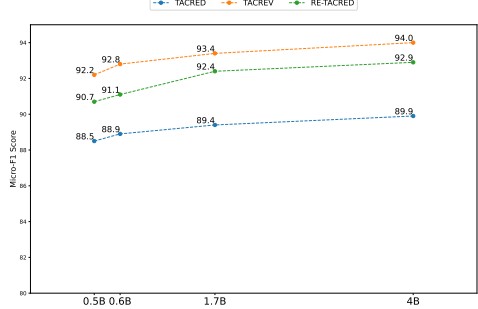

(b) Ablation Results of Model Parameters Scale

Figure 3: **Ablation studies.** (a) Component ablation of PLP-RC. Removing Line or Plane information consistently decreases performance, highlighting the complementary role of each component based on Qwen3-1.7B. (b) Model scaling ablation. Increasing the model size from 0.5B to 4B parameters steadily improves results on TACRED, TACREV, and RE-TACRED, although the gains diminish at larger scales.

**Ablation of computational efficiency**: To further examine the practicality of our approach, we conduct an ablation study on computational efficiency with 1.7B model on TACRED. As shown in Table 2, the baseline model (w/o Line and Plane) requires 16 minutes per epoch on 1*H100 GPU. Introducing the Line component increases the time to 24.2 minutes, while the full PLP-RC model requires 25.5 minutes. Despite incurring moderate computational overhead, the method offers consistent performance gains with negligible impact on overall training cost, making it suitable for large-scale applications where accuracy is prioritized. Moreover, given the relatively short training time per epoch, the overall training cost does not increase noticeably.

| Method | Time (min) |
|---|---|
| Point–Line–Plane | 25.5 |
| -w/o Line | 24.2 |
| -w/o Line and Plane | 16.0 |

Table 2: Efficiency Comparison

**Ablation of position of the instruction:** There exist many instruction-enhanced embedding methods, such as BGE and GTE, whose primary goal is to improve the representation capability of the model and its adaptability to downstream tasks. Instruction concatenation is commonly performed either at the beginning or at the end of the input sequence. We additionally conducted an ablation study on the effect of instruction position on 4*A100 GPU. The experimental results are shown in Table 3. The results indicate that prefix and suffix prompting yield nearly identical performance, with no significant difference across the three datasets.

| Pos | TACRED | TACREV | RE-TACRED |
|---|---|---|---|
| Prefix | 89.6 | 93.4 | 92.1 |
| Suffix | 89.4 | 93.5 | 92.2 |

Table 3: Ablation of instruction position

**Ablation of comparison with the generative method:**

To further distinguish whether the performance gains come from the pretrained model itself or from our proposed PLP method, we compared our approach with a generative baseline and conducted additional analyzes. We further evaluate a generative modeling setting on Qwen3 series using both zero-shot generation and supervised fine-tuning (SFT) on RE-TACRED.

In the zero-shot setting, using only a simple instruction cannot reliably enforce instruction following. The model often generates chain-of-thought (COT) (Yu et al. (2025)) reasoning instead of directly producing the target relation label. We attribute this behavior to the inherent generative preference shaped during pre-training, where the model was mainly exposed to long-form open-ended generation rather than discriminative relation classification supervision.

Therefore, we conducted SFT training based on the MS-Swift framework, and the performance on the test set is as follows in Table 4.

| Model | Zero-Shot | SFT |
|---|---|---|
| Qwen3-1.7B | No IF | 45.6 |
| Qwen3-4B | No IF | 57.9 |

Table 4: Generation Comparison

These results demonstrate the advantages of our discriminative modeling paradigm. Although generative models combined with RAG (Arslan et al. (2024)) or reinforcement learning (DeepSeek-AI (2025)) may further improve performance, the SFT results suggest that they still do not surpass our PLP method. Additionally, generative approaches have computational inefficiencies. Our smaller model thus achieves better performance with significantly higher computational efficiency than generative alternatives.

## 5.6 ANALYSIS

We further analyze the behavior of PLP-RC from multiple perspectives. First, we analyze how the model handles long-context inputs, particularly problems involving cross-sentences. Our architecture naturally supports long-context inputs. Since LLMs can encode extremely long text spans (e.g., Qwen3-4B supports up to 40k tokens), the discriminative framework offers a clear advantage over previous BERT models that have limited context window. Building on the LLM backbone, our approach is able to represent cross-sentence long-range context more effectively. Second, we analyze whether certain types of relations have a disproportionate impact on overall performance on the 1.7B model on RE-TACRED by computing the F1 score for each relation type. The full results are provided in the appendix. Notably, the underperforming categories share common characteristics in the test set: they all contain relatively few samples. This extreme data imbalance may have led to the lower performance observed for these specific categories. Future improvements could focus on addressing the class imbalance, which may further enhance the performance of our method. Additionally, our task follows a pretrain-finetune paradigm, where the LLM is used as an encoder for downstream tasks, avoiding the generative process and thereby mitigating hallucination issues.

## 6 CONCLUSION

This paper presents the Point-Line-Plane fusion method for relation classification(PLP-RC), a span-based discriminative framework that builds on LLM embeddings. The experiments demonstrate that Point-Line-Plane substantially improves relation classification, achieving SOTA results even with relatively small decoder-only LLM backbones. Ablation studies show that Plane and Instruction contribute the most, while Line provides complementary benefits, confirming the necessity of all components. Scaling experiments further indicate consistent gains with larger models but reveal diminishing returns as parameter size grows. Importantly, PLP-RC introduces negligible computational overhead, yielding a favorable trade-off between accuracy and efficiency. Ablation experiments comparing our approach with the SFT generative paradigm demonstrate that the observed advantages primarily stem from the proposed method itself. Unlike recent generative approaches to relation extraction, our framework adopts a discriminative formulation, which not only leverages LLM representations more effectively but also avoids the hallucination issues often observed in generative models. Moreover, unlike discriminative models like BERT, our use of a LLM as the encoder inherently enables processing of extremely long contexts. Overall, these results establish PLP-RC as a robust and scalable representation framework for LLM-based relation classification.

## 7 LIMITATION

While our proposed PLP-RC representation enhances entity-level reasoning and achieves consistent improvements across benchmarks, several limitations remain. First, the majority of our performance gains stem from the proposed PLP-RC method, with additional contributions from the underlying pretrained model, whose representational capacity ultimately bounds the overall performance. Second, although our method is effective even with smaller models, its computational efficiency still has room for improvement. Third, our evaluation does not fully reflect cross-sentence relational reasoning. However, since all existing relation classification datasets are sentence-level only, systematic evaluation of cross-sentence cases is not currently available. Addressing these issues presents a valuable foundation for future research.

## REPRODUCIBILITY STATEMENT

To ensure the reproducibility of our experiments, we provide the complete source code and datasets as supplementary material. Our implementation allows training models starting from the pre-trained language models. Based on the training scripts, dataset, and hyperparameters we provide in the appendix, researchers can easily reproduce our results.

## ETHICS STATEMENT

We adhere to responsible research principles, using only publicly available datasets and ensuring ethical data usage. We acknowledge potential biases in the data and model outputs, and we encourage careful and responsible application of our methods to avoid societal harm.

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

## A APPENDIX

### A.1 LLMS STATEMENT

Since we are not native English speakers, we leverage LLMs for English refinement in our paper.

### A.2 DATASET DETAILS

All the data we use are open-source and can be accessed from the Internet and we can provide the preprocessed datasets. The splits and sizes of each dataset are as follows in Table 5.

Table 5: Overview of benchmark datasets.

| Split | TACRED | TACREV | Re-TACRED |
|---|---|---|---|
| Training | 68124 | 68124 | 58465 |
| Test | 15509 | 15509 | 13418 |
| Validation | 22631 | 22631 | 19584 |
| Number of Relations | 42 | 42 | 40 |

## A.3 HYPERPARAMETER SETTINGS

In this section, we provide the detailed hyperparameter settings used in our experiments. These settings are chosen based on preliminary tuning on the validation set. Specifically, our 1.7B model on TACRED is trained with the hyperparameters listed in Table 6, while additional configurations can be found in the released code.

Table 6: Hyperparameter settings

| Hyperparameter | Value | Hyperparameter | Value |
|---|---|---|---|
| Learning rate | $3 \times 10^{-5}$ | Learning rate scheduler | constant |
| Batch size | 24 | Weight decay | 0.01 |
| Number of epochs | 10 | Warmup steps | 1000 |
| Optimizer | AdamW | Random seed | 42 |

## A.4 F1 SCORES FOR SPECIFIC RELATION TYPES

Analyzing performance across specific relation types can indeed help us quickly identify potential weaknesses of the model. Therefore, we conducted a per-category metric analysis based on the results of the 1.7B model on Re-TACRED. It is worth noting that the newly added ablation experiments were conducted in slightly different environments — including GPU models and CUDA versions — which may result in minor fluctuations in the reported metrics.The results are as follows in 7

Table 7: F1 scores for different relation labels (two-column compact format)

| Label | Count | F1 | Label | Count | F1 |
|---|---|---|---|---|---|
| per:parents | 106 | 0.90 | org:alternate_names | 337 | 0.96 |
| per:siblings | 66 | 0.95 | org:city_of_branch | 129 | 0.83 |
| per:stateorprovince_of_death | 16 | 0.32 | no_relation | 7770 | 0.94 |
| per:children | 55 | 0.81 | org:country_of_branch | 166 | 0.86 |
| org:stateorprovince_of_branch | 57 | 0.88 | per:stateorprovince_of_birth | 9 | 0.89 |
| per:city_of_birth | 15 | 0.72 | per:charges | 126 | 0.85 |
| per:country_of_death | 14 | 0.13 | per:identity | 2036 | 0.95 |
| per:country_of_birth | 0 | 0.00 | per:age | 208 | 0.98 |
| org:dissolved | 5 | 0.00 | org:founded_by | 84 | 0.89 |
| per:cities_of_residence | 125 | 0.84 | org:number_of_employees/members | 13 | 0.82 |
| per:date_of_death | 63 | 0.80 | org:founded | 34 | 0.94 |
| per:countries_of_residence | 148 | 0.69 | per:other_family | 52 | 0.95 |
| per:city_of_death | 26 | 0.36 | per:date_of_birth | 7 | 0.82 |
| org:members | 63 | 0.71 | per:cause_of_death | 50 | 0.84 |
| per:title | 523 | 0.94 | per:schools_attended | 33 | 0.86 |
| per:employee_of | 332 | 0.85 | org:top_members/employees | 295 | 0.91 |
| per:stateorprovinces_of_residence | 73 | 0.78 | org:shareholders | 12 | 0.43 |
| org:political/religious_affiliation | 29 | 0.81 | per:origin | 115 | 0.84 |
| org:website | 30 | 0.76 | per:spouse | 73 | 0.91 |
| per:religion | 59 | 0.68 | org:member_of | 64 | 0.63 |