# OpenReview forum: "PLP-RC:Point–Line–Plane Fusion for Discriminative Relation Classification with LLMs"
_ICLR.cc/2026/Conference — Submitted to ICLR 2026_

### Official Review · Reviewer_zsse · 2025-10-21

**Soundness:** 3
**Presentation:** 3
**Contribution:** 3
**Rating:** 8
**Confidence:** 4

**Summary:**

This paper proposes PLP-RC, a framework for Relation Classification that leverages Large Language Models (LLMs) embeddings but avoids the hallucination issues common in LLM generative usages. The core innovation is a "Point-Line-Plane" geometric fusion mechanism: Point represents local entity span information, Plane encodes global context  with the [EOS] token, and Line (attention scores between entities and [EOS]) bridges the two levels of granularity. These features are integrated into fused entity representations for relation prediction. PLP-RC achieves new state-of-the-art results on the TACRED, TACREV, and RE-TACRED benchmarks, outperforming both discriminative and generative baselines.

**Strengths:**

* The paper proposes a discriminative approach with LLM embeddings instead of directly using the LLM generative power
* The Point-Line-Plane geometric analogy for feature fusion is interesting
* The experimental evaluation is comprehensive, including benchmarks, baselines, ablation studies, model scaling and computational costs
* The paper is well-written and clear

**Weaknesses:**

* The evaluation benchmarks primarily consist of single-sentence contexts. The authors may briefly discuss the proposed approach's adaptation to long-range dependencies.
* The ablation studies show that the "Line" component's contribution is relatively modest compared with others.
* It would be nice if the authors could analyze further whether specific types of relations affect the overall performance

**Questions:**

1. Given the most contribution from "Plane" and instruction, how does the model's performance change if the instruction changes, e.g., placed at the beginning instead of the end?
2. The "Line" feature uses attention scores from the [EOS] token to entity tokens. Did you try other attention scores like those between the subject and object entities?

---

> ### Author Response · Authors · 2025-11-19
> **Thanks for Review:**
>
> **We sincerely thank the reviewer for the valuable time and careful consideration dedicated to our manuscript. The insightful comments and constructive suggestions provided by the reviewer are of great significance for improving our work, and we would like to express our heartfelt gratitude to you for your recognition and support.**
>
> **Response:**
>
> **Q1:The evaluation benchmarks are primarily based on single-sentence contexts. Please explain how the method adapts to long-distance dependencies.**
>
> We acknowledge that studying long cross-sentence contexts is important. It is true that our experiments were conducted only on sentence-level benchmarks, and we did not evaluate the model’s capability in cross-sentence relation classification. The main reason is that current benchmarks for relation classification are still sentence-level, and recent work—including prior studies—has also relied on TACRED-series datasets.
>
> Nevertheless, our architecture is inherently capable of handling long contexts. Since LLMs can encode extremely long input sequences (e.g., Qwen3-4B supports up to 40k tokens), it offers a substantial advantage over previous BERT-base models that are limited to 512 tokens. Thus, with our LLM-based design, the framework is able to represent cross-sentence and ultra-long contexts more effectively.
>
> For the long cross-sentence setting, we have conducted preliminary experiments and analysis on a private dataset. Instead of using the original [EOS] token, we select tokens located at a certain distance after the entity pair as a global compressed representation, yielding a more precise encoding of contextual information. However, due to the lack of publicly available benchmarks for cross-sentence relation classification, we currently do not have widely verifiable results. This is a limitation of our work.
> We therefore explicitly list cross-sentence relation classification as an important direction for future development of our framework.
>
> **Q2: The contribution of line-level information seems relatively small.**
>
> Indeed, the effect of line-level information is weaker compared to plane-level information and instruction prompting. However, its contribution should not be underestimated. The motivation for introducing line information is twofold: first, plane information alone does not fully capture contextual representation; second, line information enables modeling the interaction between entities and their surrounding context. Guided by this motivation, we designed the line-level representation.
>
> The experimental results show that adding line information only incurs a very small computational overhead, while consistently improving the PLP-RC performance metrics. This demonstrates the effectiveness of incorporating line information.
>
> **Q3: Do specific relation types affect overall performance?**
>
> This is an excellent question. Analyzing performance across specific relation types can indeed help us quickly identify potential weaknesses of the model. Therefore, we conducted a per-category metric analysis based on the results of the 1.7B model on Re-TACRED. The sub results are as follows, and the full results can be found in the appendix.
>
> label                     | f1    | count
> --------------------------|-------|------
> per:parents               | 0.90  | 106
> per:siblings              | 0.95  | 66
> per:stateorprovince_of_death | 0.32  | 16
> org:alternate_names       | 0.96  | 337
> org:city_of_branch        | 0.83  | 129
> no_relation               | 0.94  | 7770
>
> The poorly performing categories share some common characteristics in the test set — they each contain relatively few samples. This extreme data imbalance likely leads to lower metrics for those individual categories. Future improvements can focus on addressing class imbalance to further enhance our method.
>
> **Q4: How will the model's performance behave if the position of the instructions changes?**
>
> There exist many instruction-enhanced embedding methods,whose primary goal is to improve the representation capability. In current practice, instruction concatenation is commonly performed either at the beginning or at the end of the input sequence.
> To address the reviewer’s concern, we additionally conducted an ablation study on the effect of instruction position. The experimental results are shown below:
>
> Method        | Tacred | Tacrev | Re-tacred
> --------------|--------|--------|----------
> Prefix Prompt | 89.6   | 93.4   | 92.1
> Suffix Prompt | 89.4   | 93.5   | 92.2
>
> The results indicate that prefix and suffix prompting yield nearly identical performance, with no significant difference across the three datasets.
>
> **Q5：Alternative ways to implement line information:**
>
> We appreciate the reviewer’s suggestion and agree it is promising; however, due to time constraints, we could not conduct the corresponding experiments during the rebuttal stage and plan to explore this idea more systematically in future work.

---

### Official Review · Reviewer_4Tdp · 2025-10-26

**Soundness:** 3
**Presentation:** 2
**Contribution:** 3
**Rating:** 4
**Confidence:** 4

**Summary:**

The paper presents a novel framework, Point–Line–Plane Fusion (PLPF), for relation classification using Large Language Model (LLM) embeddings. The work addresses a long-standing challenge in balancing local entity representation with global contextual understanding while mitigating hallucination issues common in generative models. The proposed geometric abstraction models entities as points, context alignment as lines, and overall semantic scope as planes, which is conceptually elegant and empirically validated.

**Strengths:**

1.The paper introduces an innovative geometric fusion paradigm based on the point–line–plane concept, which offers a clear and interpretable approach to integrating local and global features for relation classification.

2.The proposed method leverages Large Language Model embeddings in a discriminative framework, effectively mitigating hallucination issues that are commonly observed in generative LLM applications.

**Weaknesses:**

1.The core "Point-Line-Plane (PLP)" fusion mechanism lacks sufficient theoretical justification. The paper frames the mechanism as "conceptually grounded in geometric and information-theoretic principles" but provides no formal connection to these principles (e.g., how line/plane representations map to information-theoretic metrics like mutual information).

2.The methodological description lacks mathematical rigor and theoretical foundation. The "geometric perspective" remains largely metaphorical without formal mathematical formulation or theoretical guarantees about the representation properties.

3.The paper fails to address cross-sentence relation classification, a critical limitation of existing methods highlighted in the Introduction. All experiments are conducted on sentence-level datasets, yet the PLP framework is claimed to "capture long-range dependencies"—no evidence is provided for this capability, and the [EOS] token’s causal attention (autoregressive) cannot model cross-sentence context effectively.

4.The paper confuses discriminative vs. generative paradigms. It claims PLP-RC avoids hallucinations by using a "discriminative framework" but uses decoder-only LLMs (Qwen3) pretrained with generative next-token prediction.

**Questions:**

See Weaknesses.

---

> ### Author Response · Authors · 2025-11-19
> **Thanks for Review:**
>
> **Thanks a lot for your time in reviewing and insightful comments.**
>
> **Response:**
>
> **Q1: The  PLP design lacks theoretical foundations. The geometric perspective and information-theoretic analogy lack strict mathematical rigor.**
>
> Thank you for the valuable comment. The “point–line–plane ” naming is  a conceptual analogy intended to intuitively illustrate the complementary roles of three different types of information sources:
> - Point (span token): the pretrained representation of a single token, which captures local semantics but lacks global context;
> - Line (attention score): attention scores describe the inter-token association, compensating for the missing structural information in simple weighted aggregation;
> - Plane (global contextual token, e.g., [EOS] ): in causal modeling of LLMs, information gradually converges toward later positions, enabling later tokens to encode more holistic contextual semantics.
>
> This naming is not meant to imply a strict derivation from geometry or information theory. A more theoretical interpretation of PLP is as follows: due to causal attention in LLMs , each token can only attend to preceding tokens and cannot directly observe future tokens; meanwhile, the next-token-prediction pretraining objective focuses on generating the next token rather than encoding precise bidirectional context, leading to potentially incomplete information. Motivated by this, we conducted ablation studies showing that incorporating both line-level and plane-level information yields further performance gains, confirming that the fused PLP representation provides more comprehensive contextual information. As the reviewer correctly pointed out, the “geometric perspective” in PLP is intended for intuitive understanding rather than for expressing mathematically rigorous derivations. We will soften and clarify this part in the revised manuscript.
>
> **Q2:  Concern about "the method does not address cross sentence issues and the EOS  cannot effectively represent cross-sentence information".**
>
> We sincerely appreciate this insightful observation. We fully agree on the importance of inter-sentence relation classification and would like to clarify the following: our architecture leverages LLMs (Qwen3 as encoders, which support ultra-long contexts and are therefore structurally capable of handling inter-sentence relational reasoning. This contrasts with prior discriminative approaches that primarily rely on BERT-style encoders with limited input lengths, which is one of the intrinsic advantages of our framework.
>
> In the datasets used in our experiments, sentence lengths tend to be short. Under causal attention (where later tokens can attend to all previous tokens), the [EOS] token can effectively aggregate preceding contextual information. Therefore, we choose [EOS] as the global compressed representation. For long-context cross-sentence cases, we conducted preliminary experiments on the private dataset, where we replaced [EOS] with a token selected at a fixed distance after the entity pair. This allows capturing context more precisely. However, more comprehensive experiments are needed to fully validate this approach.
>
> We concur that under current sentence-level benchmarking, the evaluation does not fully reflect cross-sentence relational reasoning. However, since all existing relation classification datasets, including those used in this submission, are sentence-level only, systematic evaluation of cross-sentence cases is not currently avaliable. We explicitly list cross-sentence relation classification as an important future direction for our framework.
>
> **Q3：The distinction between generative and discriminative paradigms is confused.**
>
> We thank the reviewer for this insightful comment. In our paper, the generative vs. discriminative distinction refers to paradigms of downstream tasks rather than to the pre-training stage. In our PLP-RC framework, the LLM does not leverage any generative capability in the downstream task. Instead, a decoder-only LLM is used purely as a feature extractor/encoder, thereby avoiding the autoregressive generation process during task inference and eliminating hallucinations in downstream prediction.

---

### Official Review · Reviewer_UvcE · 2025-10-28

**Soundness:** 2
**Presentation:** 2
**Contribution:** 2
**Rating:** 4
**Confidence:** 4

**Summary:**

This paper proposes a Point-Line-Plant fusion framework based on LLM embeddings for entity relation classification, which leverages the representational capacity of LLMs while mitigating their hallucination issues. Experiments were conducted to validate the effectiveness of the proposed method.

**Strengths:**

The proposed approach utilizes a LLM as text encoder, effectively transferring its rich representational capacity to the discriminative task of relation classification. This enables the generation of semantically richer embeddings, leading to improved classification performance. Furthermore, the method requires no fine-tuning of the LLM, thereby avoiding substantial computational costs.

**Weaknesses:**

1. Lines 52-54 of the paper mention that LLMs have inherent deficiencies in capturing contextual content, yet there is no further explanation or citation of relevant arguments in the paper. Taking the Qwen3 model used in the paper as an example, it can support a maximum context length of 128K, which is fully capable of covering some basic tasks including Relation Classification.

2. Experiments demonstrate that the PLP-RC method proposed in this paper is effective in relation classification, significantly outperforming other approaches. However, PLP-RC uses Qwen3 as its backbone, while the comparison methods adopt GLM-10B, Mistral-7B, and LlaMA2-7B. It is important to note that Qwen3 is a new-generation model; its 4B version even outperforms Qwen2.5-7B, its predecessor. Moreover, Qwen2.5-7B itself shows better performance than models like Mistral-7B and LlaMA2-7B. Therefore, it remains unclear whether the advantage of PLP-RC over other models stems from the method itself or from Qwen3, resulting in a lack of fair comparison.

3. As a mature and fundamental task, relation classification can already achieve good results by directly using LLMs to generate answers. PLP-RC treats LLMs as encoders and transforms relation classification into a discriminative task, but the paper lacks experiments to illustrate the advantages of PLP-RC compared to direct answer generation.

4. The writing expression of the paper needs further polishing, and the presentation should be consistent throughout the text. Some parts of the article lack necessary citations, such as the reference to the strategies of previous work in lines 222-223.

**Questions:**

Refer to the weakness.

---

> ### Author Response · Authors · 2025-11-19
> **Thanks for Review:**
>
> **The comments provided by the reviewers are very useful for our work.**
>
> **Response:**
>
> **Q1: Explaination of  Pretrain Language Models have Inherent deficiencies in capturing contextual content**
>
> The goal of PLP is to enhance the ability of embeddings to capture global semantic information. In both BERT-style models (Masked Token Prediction) and  LLMs ( Next-Token Prediction), the pretrained representation of each token is primarily optimized for the pretraining objective rather than for encoding the full contextual semantics. Moreover, in LLMs with causal attention, each token can only attend to its preceding tokens but not to future ones, which inherently limits its ability to encode complete contextual meaning. This insufficient contextual expressiveness at the token level is exactly the challenge we want to address. Although Qwen3 supports very long context, it does not  resolve the problem that a single token still represents limited global semantic information.
>
> **Q2,Q3: Uncertainty remains as to whether the performance gain comes from Qwen3 or from PLP itself, and no comparison is provided with a generation method**
>
> Thank you for the reviewer’s valuable comments. We fully agree that capability gaps across LLMs may affect the fairness of comparison, and that the absence of direct comparison with generative approaches could be a concern. To address this, we have added new ablation studies as follows.
>
> **1. Discriminative baseline using Qwen embeddings (without PLP and without instruction prompts)**
>
> We evaluate a baseline that uses Qwen3 embeddings with a span-based discriminative classifier, without the proposed Point–Line–Plane (PLP) paradigm and without instruction prompts. The results are as follows:
>
> | Model     | Tacred | Tacrev | Re-Tacred |
> |:----------|:-------|:-------|:----------|
> | PLP-RC    | 89.4   | 93.4   | 92.4      |
> | Span Only | 85.3   | 88.4   | 82.8      |
>
> Compared to the original span-only approach, adding PLP and instruction prompts yields substantial performance gains across all datasets. Meanwhile, the span-only baseline still performs reasonably well, demonstrating that the Qwen model itself has strong representation ability, even without our method.
>
> **2. Comparison with generative method**
>
> We further evaluate a generative modeling setting on Qwen3 using both zero-shot prompting and supervised fine-tuning (SFT):
> - In the zero-shot setting, the model exhibits severe instruction-following failure, making it unable to directly output the relation label as requested.
> - With SFT using the MS-SWIFT training pipeline, Qwen3-4B achieves a Micro-F1 of 57.91, which is lower than our discriminative method, and also lower than RAG-based generative approaches with Mistral-7B and Llama2-7B reported in literature.
>
> | Model       | Zero shot | SFT  |
> |:-----------|:---------|:----|
> | Qwen3-1.7B | No Instruction Following | 45.6 |
> | Qwen3-4B   | No Instruction Following | 57.9 |
>
> These results demonstrate the advantage of our discriminative modeling paradigm. Although generative models combined with RAG or reinforcement learning may further improve performance, the SFT results suggest that they still do not surpass our method. Additionally, generative approaches have computational inefficiencies. Our smaller model thus achieves better performance with significantly higher computational efficiency than generative alternatives.
>
> **3. Model scaling analysis**
>
> Regarding scaling, generative capability indeed improves across model generations and with larger parameter sizes. However, the performance gain from using larger models is limited under the discriminative setting, and does not align with trends observed in Qwen3 Technical Report. This suggests that pure generative capability does not fully reflect effectiveness on non-generative downstream tasks such as relation classification.
>
> **Q4: Improve Writing and Add references**
>
> We thank the reviewer for the reminder. We have checked the entire manuscript and unified expressions that were inconsistent or imprecise. These revisions include, but are not limited to: standardizing “EOS” as “[EOS]”, “SpanBERT” as “Span Bert”, and modifying “RAG4RE” to “RAGRE”, among others. And we have added the reference.

---

> > ### Comment · Reviewer_UvcE · 2025-11-28
> >
> > Thank you for your responses. I appreciate the authors’ efforts to address my comments. However, the replies do not resolve my original concerns, and the paper still lacks clear and substantive innovation. Therefore, I will maintain my current score.

---

> > > ### Author Response · Authors · 2025-11-28
> > > **Thanks for Additional Feedback:**
> > >
> > > Dear Reviewer UvcE,
> > >
> > > Thank you again for your time and for the additional feedback. We respect your review and understand your concerns about our current presentation of the contributions.
> > >
> > > To help us further improve the paper, could you kindly clarify which of your original concerns you believe remain unresolved? We would sincerely appreciate the specific guidance .
> > >
> > > Thank you again for your constructive comments, and we truly value your feedback.
> > >
> > > Best regards,
> > >
> > > Authors

---

### Official Review · Reviewer_v33w · 2025-10-31

**Soundness:** 1
**Presentation:** 1
**Contribution:** 1
**Rating:** 0
**Confidence:** 5

**Summary:**

This paper introduces the Point–Line–Plane (PLP) framework for relation classification, demonstrating limited performance improvements on TACRED and related datasets. However, the work lacks substantial novelty and does not meet high standards for innovation and impact.

**Strengths:**

This paper would be a good negative example to warn students not to write similar papers.

**Weaknesses:**

The paper lacks novelty, practical application, and sufficient experimental rigor. To improve, the authors should focus on contemporary challenges, explore more innovative approaches, and validate real-world applicability.

**Questions:**

Suggestion：The authors should consider more research-worthy directions.

---

> ### Author Response · Authors · 2025-11-19
> **Thanks for Review**
>
> **Reply:**
>
> **Q1： Paper lacks novelty, practical application, and experimental rigor.**
>
>  Thank you very much for your time and feedback. We would like to clarify that this work makes concrete efforts regarding novelty, practical application, and experimental rigor. Our method adopts a discriminative approach combined with LLMs and achieves **SOTA performance** on the TACRED benchmark, significantly surpassing both previous discriminative methods and LLM-based generative approaches. Using a smaller-parameter model while outperforming much larger models leads to **higher computational efficiency**, and should not be regarded as merely marginal performance gains or limited innovation.
>
> In terms of practical value, our approach provides a new paradigm for representation learning, enabling a new way of solving fundamental NLP tasks. Regarding experimental rigor, we conducted multiple sets of ablation studies to analyze the contribution of each component. In addition, we investigated the scaling law characteristics of the model as well as experiments on computational efficiency.
>
> Therefore, we respectfully argue that this work presents not a minor improvement, but a contribution with practical significance and genuine methodological innovation. We kindly ask the reviewer to reconsider their evaluation of our manuscript.
>
> **Q2： Consider focusing on more valuable/ trending research directions.**
>
>  We sincerely appreciate the reviewer’s suggestion and fully understand the importance of keeping up with research trends. However, academic research is inherently diverse, rather than solely driven by what is currently most popular. We would like to emphasize that **advancing fundamental tasks with new perspectives and methodologies is equally crucial**. Major and meaningful innovations often stem from rethinking core problems using new tools and stronger frameworks，such as large language models—providing a robust, reproducible, and extensible solution.
>
> We hope our work can be recognized as a meaningful attempt to **extend the value of LLMs from trending applications to foundational NLP tasks**, contributing to the community in a sustainable and impactful manner.

---

> > ### Comment · Reviewer_v33w · 2025-11-26
> >
> > Thank the authors for their response. The overall contribution and significance of the work remain unresolved. Therefore, I will keep my current rating.

---

### Comment · Area_Chair_yvav · 2025-11-25
**Discussion Period**

Dear Reviewers and Authors,

Thank you to the authors for submitting your rebuttal. We kindly encourage reviewers to take a moment to read the response and share any follow-up thoughts. Your timely engagement at this stage is highly valuable and helps ensure a fair, well-informed final decision.

We appreciate everyone’s efforts and contributions to the process.

Warm regards,
Your AC

---

### Author Response · Authors · 2025-11-27
**Discussion due is approaching**

Dear Reviewers,

Thanks a lot for your time in reviewing and insightful comments, according to which we have revised the paper to answer the questions carefully. We sincerely understand you’re busy. But since the discussion due is approaching, would you mind checking the response to confirm where you have any further questions?

We are looking forward to your reply and happy to answer your questions.

Best regards

Authors

---

### Meta-Review · Area_Chair_VXpU · 2026-01-07

**Summary:**

Reviewers are split. One reviewer thinks the paper is far below the bar and argues the method is basically a trivial mix of existing ideas with weak writing and weak rigor. The other reviewers agree the method can work and the results look strong on TACRED-style benchmarks, but they still question what is truly new here, whether the comparisons are fair (given the backbone choice), and whether the paper actually delivers on its motivation about handling longer context beyond single sentences.

**Reviewer Concerns:**

Some points were addressed in the rebuttal. In particular, the authors added an ablation that uses Qwen embeddings with a simple span-based classifier (“span-only”) to show that PLP + prompting helps beyond just choosing a strong model. They also added a generative baseline (zero-shot failing to follow the label format and SFT being much worse than PLP-RC), which supports their claim that the discriminative setup is competitive and more efficient. They clarified smaller presentation points and showed that prefix vs suffix instructions make little difference.

However, the main concerns are still there. The “Point–Line–Plane” idea still reads more like a nice metaphor for combining span features, attention weights, and an [EOS] pooled vector than a genuinely new method with clear technical depth; even the authors admit it is not a rigorous geometric/information-theoretic theory. Also, the paper motivates long-range or cross-sentence relation reasoning, but the experiments are still on single-sentence datasets, so the claimed advantage on longer contexts is not actually demonstrated. Finally, some reviewers remain unconvinced that the “line” component matters much (its gain looks small), making it feel like most of the improvement may come from a strong LLM encoder + instruction prompting + global pooling, rather than a new framework.

**Reviewer Scores:**

I do not expect big score changes after rebuttal. Reviewer v33w is extremely negative and doubled down in a follow-up comment, so they likely stay at 0 (strong reject). Reviewer UvcE explicitly says the rebuttal did not resolve their core issues and they will keep 4 (borderline reject). Reviewer 4Tdp raised concerns about lack of theory and the mismatch between “long-range” motivation and sentence-level evaluation; the rebuttal mostly concedes these points, so they likely stay at 4. Reviewer zsse was already very positive and the added ablations support their view, so they likely remain at 8 (accept). Net: the likely post-discussion shape remains roughly same score, which is too inconsistent and too weak on novelty/claims for acceptance.

---

### Decision · Program_Chairs · 2026-01-26

Reject